## Replication

cognition

context-dependent memory, replication study, divers

**Author for correspondence:**
Jaap M. J. Murre
e-mail: jaap@murre.com

# The Godden and Baddeley (1975) experiment on context-dependent memory on land and underwater: a replication

## Jaap M. J. Murre

Department of Psychology, University of Amsterdam, PO Box 95195, 1001 NK Amsterdam, The Netherlands

JMJM, 0000-0003-4447-9482

A replication of the experiment by Godden and Baddeley (Godden and Baddeley 1975 *British Journal of Psychology* **66**, 325–331 (doi:10.1111/j.2044-8295.1975.tb01468.x)) on environmental context-dependent memory is described. Sixteen divers studied auditorily presented word lists on land or underwater and recalled these 4 min later on land or underwater (each diver participated in all four combinations). Contrary to the original study, we did not find that recall in the same context where the words had been learned was better than recall in the other context. We discuss differences between our replication and the original study and emphasize the importance of attempts at replicating classic studies.

## 1. Introduction

I have been teaching courses on *Cognitive Psychology* and *Memory and Learning* for 30 years, and virtually all textbooks I have used describe the classic study on environmental context-dependent memory by Godden & Baddeley [1], usually in some detail. These authors had divers learn spoken words, either on land or underwater, and after 4 min tested them in either environment; all divers did all four combinations of environments. In this classic study, they found a large effect of context: whether learning underwater or on land, recall of the materials was much better in the same environment.

Environmental context-dependent memory measured with free recall is now well established, although in many circumstances the effect is rather weak with an effect size (Cohen's $d$) of around 0.25 [2]. It is, therefore, perfectly feasible that the Godden & Baddeley [1] result, which also used free

recall, is robust. Environmental context-dependent *recognition* was once thought to be non-existent [3] but more recently several studies have found evidence for it and delineated some of the circumstances under which it occurs (e.g. [4]).

As is true for several other classic studies, no replication of Godden & Baddeley [1] has ever been published, although there is a study with novice divers [5] who learn how to use decompression tables in one environmental context and are better at applying these in the same environment (e.g. learned underwater, used underwater) than the other (e.g. learned underwater, used on land). Also, the Godden and Baddeley study was itself inspired by anecdotal observations on context-dependent memory underwater and on land by Alan Baddeley when working with divers using the 'Wrecks Test' [6].

To find other replication studies, we consulted one of the authors of the original study but he did not know of any replications, published or not (Alan Baddeley 2014, personal communication). It is possible that some people attempted replication and failed, losing the incentive to publish or suffering rejections by journals not inclined to publish replications, especially failures to replicate. It is also possible that no one has tried, which is understandable given the complexities of organizing an experiment with divers as subjects, as Godden and Baddeley mention several times when commenting on their procedure in the 1975 paper.

My motivation to undertake this replication is that I feel it is of great importance to try to replicate important results in psychology and even more so if studies are famous. As Jonathan Russell [7] argued in *Nature*: 'If it is a job worth doing, it is worth doing twice'. This is why we attempted to replicate Ebbinghaus' classic forgetting curve from 1885 [8,9]. This yielded comparable results to the original Ebbinghaus study and provided novel opportunities for analysis and comparison [10]. Not all classic studies are easily replicated, however, such as the classic study by Bartlett [11] on the schematization of stories from unfamiliar cultures, which until 1999 had only had unsuccessful replication attempts until Bergman & Roediger [12] succeeded in replicating the basic findings. One of the reasons earlier replications may have failed is because not all details were well documented in the original study from 1932. The exact instructions, for example, were not included. Replication of classic experiments, thus, serves the dual purpose of verifying the reliability of the original results and uncovering more precisely how the original experiment was conducted. In our Ebbinghaus replication, for example, the missing schedule of exactly at what dates the saving sessions had been scheduled made it hard to replicate the 31-day data point, because we could not reconstruct how many potentially interfering sessions had taken place between the learning and testing after 31 days.

Replication studies are more common nowadays, though when trying to submit our Ebbinghaus replication, two memory psychology journals rejected our paper outright, because it was a replication 'that did not advance our knowledge of any theory'. One reviewer did not believe 'a study is worth replicating just because it's famous'. I strongly disagree. These studies are the foundation of what we are teaching our students and if they are unreliable or wrong, it is of paramount importance to know so. This is why we undertook the following replication attempt. This article received results-blind in-principle acceptance (IPA) at Royal Society Open Science. Following IPA, the accepted Stage 1 version of the manuscript, not including results and discussion, was preregistered on the OSF (https://doi.org/10.17605/OSF.IO/KFHVU). This preregistration was performed after data analysis. The author, furthermore, declares to have no competing interests. The study was funded by the University of Amsterdam as part of my normal work as a researcher.

## 2. Methods

The experiment took place in one day in an indoor, heated swimming pool with a depth of 180 cm that was part of an apartment building in Amsterdam-Zuid and which was made available for exclusive use by us. This was a departure from the Godden & Baddeley [1] experiment, which used open water (though two subjects were tested in an indoor pool later). The experiment was filmed for a Dutch TV programme, called *Katja's Bodyscan*, which featured the Dutch actress Katja Schuurman. She also took part in the experiment as an additional subject (in Experimental Group 3), but her data were not analysed as she also did brief narratives on camera during the experiment so that viewers were aware of what was happening. This would probably have interfered with her memory performance. Though the TV crew and filming, which took part throughout the day, may have been a distraction, the experiment itself was taken seriously by all people involved and several crew members helped to

collect and score materials. Ethical approval was obtained from the Medical-Ethical Committee of the University of Amsterdam (title in the registry: 'Geheugen Onder Water').

## 2.1. Subjects

Similar to the original study by Godden & Baddeley [1], we recruited sixteen divers (average age 29 years 6 months; 10 men, 6 women) from diving clubs in the Amsterdam area. A difference with the original study may be that their Experiment 1 used undergraduates who were amateur divers, and it is likely that their sample was more homogeneous. They were compensated for their travel expenses but did not receive a fee for participating. They all had several years of diving experience.

We could have included more subjects, for example, 32, doubling the size of each cell in the design. The effect size of the interaction effect in the original study, however, was very large. Focusing on the $F$-value of the interaction term, $F = 22.0$, we first estimate $\eta^2 = F/(F + df2)$, where $df2$ is $16 - 4 = 12$ [13], which gives $22.0/(22.0 + 12) = 0.647$. From this, we derive a lower bound for Cohen's $d = 2 \times (\eta^2/[1 - \eta^2])^{1/2}$ [14], giving $d = 2.708$, which is considered extremely large ($d = 0.8$ is normally considered 'large'). This is not surprising, given that the $p$-value is $p < 0.001$ in the original paper. Thus, replicating the original design exactly as is, should yield more than 99.9% statistical power assuming the lower bound effect size is accurate. Hence, there is a very low chance of finding a spurious null result. Given the strength of the original effect and the added complexities involved in recruiting 32 divers, we opted for an exact replication.

## 2.2. Stimuli

We selected 180 two-syllable Dutch words from the CELEX word corpus [15] that had a frequency of 26 to 84 per million words and were 5 to 6 letters in length. The words were randomly assigned to five 36-word lists: a practice list and four experimental lists.

## 2.3. Apparatus

We tested four divers at a time using an underwater speaker system, which was constantly monitored for audibility underwater. At all times, the sound could clearly be heard underwater. In the Wet condition, each diver was given a plastic pad and a pencil to write down any words they remembered during the recall phase.

## 2.4. Procedure

We aimed to mimic the procedure by Godden & Baddeley [1] as closely as possible. Due to practical constraints, we had to test all divers in a single day, rather than in 4 days. Divers arrived in groups of four in 2 h intervals. All four divers took part in an experimental session simultaneously. The first half an hour was filled with general orientation and putting on wetsuits. The remaining 1.5 h was spent on the actual experiment.

First, general information was provided about the experiment and informed consent was obtained. Following this (still on dry land), instructions were given on how to breathe with the presentation pattern (see below) so that all words could be heard clearly underwater. The divers also practised with the plastic writing pads and pencils. Then, each group put on SCUBA gear and tested it in the water. This means that the divers themselves were wet at the beginning of each of the four conditions. While they were underwater, two test runs with the underwater speakers were run, the second of which consisted of the experimenter (J.M.J.M.) reading a list of 36 words similar to the experimental lists. An experienced, licensed instructor in SCUBA gear was present underwater whenever the divers were underwater. She monitored whether underwater sounds were clearly audible (which was always the case) and made sure the divers were safe and comfortable. At all times, the experimenter was on shore to oversee the experiment and answer questions about the procedure. Between the four conditions, there was a brief rest and afterward a debriefing session in which any questions they had were answered.

The experimental lists were read by the experimenter (beforehand) at a speed of one word per 2 s in groups of three with 4 s between triplets and recorded as four MP3 files, which were used for stimulus presentation. To help the divers attain a comfortable breathing pattern, each list was preceded by reading three blocks of three $z$'s (Dutch: 'zet') in the same pattern as the experimental words followed by the

**Table 1.** Graeco-Latin design of four groups of four divers and the order and list pairings. D = Dry, W = Wet and the number refers to the MP3 file.

| | order of conditions and lists pairings | | | |
|---|---|---|---|---|
| Group 1 | DD,1 | DW,2 | WD,3 | WW,4 |
| Group 2 | WD,4 | WW,3 | DD,2 | DW,1 |
| Group 3 | WW,2 | WD,1 | DW,4 | DD,3 |
| Group 4 | DW,3 | DD,4 | WW,1 | WD,2 |

word 'breathe' (Dutch: 'ademen'). After a 10 s pause, this entire list was repeated, including the breathing pattern. After this repetition, another 10 s pause followed, after which 15 random digits were read in the same pattern as before to eliminate primary memory effects. These had to be copied immediately on the writing pads. A full presentation on an MP3 file lasted nearly 5 min.

Subjects heard the lists either above (Dry) or underwater (Wet). In the Dry condition, they were seated at the edge of the pool. Stimulus presentation was followed by the instruction to move to the other environment, which involved either removing or putting on their masks and breathing tubes. After a 4 min delay (measured from the end of the last digit), the subjects had to move back to the original environment, and once they had arrived there, they had 2 min to write down any words they could recall in any order. The end of the session was announced through the speakers (above or underwater). After a condition had ended, writing pads were collected, photographed, and scored immediately by assistants. The scores were entered into an Excel sheet.

The pairings of the four lists (i.e. MP3 files) and four conditions followed a Graeco-Latin square as in Godden & Baddeley [1], table 1.

## 3. Results

A two-way analysis of variance of average words recalled showed that there was a main effect of learning environment ($F = 16.2$, $df1 = 1$, $df2 = 60$, $p < 0.001$), no main effect of recall environment ($F = 0.601$, $df1 = 1$, $df2 = 60$, $p = 0.440$), and crucially no interaction effect between learning and recall environment ($F = 1.999$, $df1 = 1$, $df2 = 60$, $p = 0.163$). The data are publicly available at https://osf.io/q2vjk/ in Excel format. The main results are shown in figure 1*a* and the original results by Godden & Baddeley [1] in figure 1*b* to facilitate comparison. We used a Wilcoxon matched pairs, signed-ranks test to do detailed comparisons. When learning on land, there was no statistical difference between recall on land or underwater ($p = 0.47$, two-tailed). When learning underwater, however, recall on land was better than underwater ($p < 0.01$, two-tailed). There was no difference between learning and recalling in the same or in a different context ($p = 0.215$).

## 4. Discussion

Our aim was to replicate the original experiment by Godden & Baddeley [1] but we failed to achieve this: we did not obtain a statistically significant effect of same versus different context. Our replication was a combination of Godden and Baddeley's Experiments 1 and 2. In Experiment 1, the divers in the Dry–Dry and Wet–Wet condition were waiting quietly for 4 min, whereas we had the divers leave their environment and then return before testing. This mimics the condition of Experiment 2 in Godden & Baddeley [1]. They introduced it to limit rehearsing in the unfilled interval and to exclude (lack of) change of context as an explanation [16].

If we compare figure 1*a,b*, we see that our results were quite different from Godden & Baddeley [1] in two respects. (i) Our scores were lower on the whole, which may have been due to our using more difficult words or weaker memory in our subjects who were probably older and less well educated than in the original. (ii) In our results, it becomes clear that learning on land gave higher recall scores than learning in the water, irrespective of recall environment. This may have been caused by the underwater speaker system, although a diving instructor monitored the sound quality at all times and reported the stimuli to be clearly audible.

If we compare words learned on land and recalled on land versus underwater, we see no statistical difference. For words learned underwater, recall on land is, in fact, better and this difference was

(*a*)

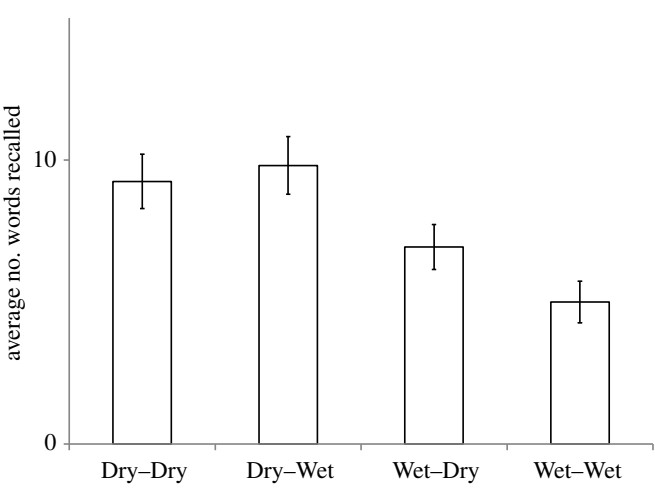

(*b*)

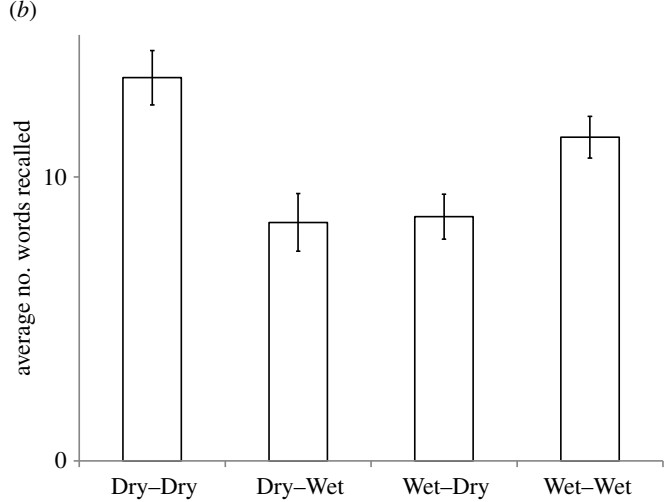

**Figure 1.** The average number of words correct with standard errors for the four conditions (table 1). (*a*) Our own data. (*b*) Data by Godden & Baddeley [1].

statistically significant. Our results, thus, not only fail to replicate Godden & Baddeley [1] but do not even have a trend in the same direction. Though we tried to mimic the original stimuli and procedure as much as possible, there were a number of important differences.

A large procedural difference was that the Godden and Baddeley divers took 4 days to do all four conditions, whereas we had one experimental group do all four conditions in about 1.5 h, which may have caused extra forgetting due to proactive interference from lists learned earlier. Another difference was that our experiment was being filmed as part of a TV programme, and this may have distracted the participants. Godden and Baddeley, however, also report many distractions to the divers during their 4 days of open Wet diving near Oban, Scotland, including almost being run over by an ex-army DUKW. Apart from that, a large difference in general conditions is probably the use of a heated indoor pool, which was about 180 cm deep instead of the open water dive near Oban, where the divers did the underwater part of the experiment at a reported 7 m depth in cold water (except for two divers who were later replaced and used an indoor pool as well). One could argue that the difference in context is much larger for a deep open water dive than for a shallower indoor pool dive. Also, it is possible that at greater depth, the physiology of a person changes due to much higher pressure. If that would be the case, it would be evidence not so much for context-dependent memory but for state-dependent memory.

The Dry–Dry and Wet–Wet conditions each included *two* context changes rather than one in the Dry–Wet and Wet–Dry conditions, which may have obscured context-dependent memory. Also, if the findings of Godden and Baddeley's Experiment 2 hold, the effect of context changes as such would

not have altered our main conclusions, as they found waiting out of context (i.e. Dry–Wet–Dry) to be comparable to waiting in the same context (Dry–Dry–Dry).

We have to admit that we were disappointed by our failure to replicate. Getting a group of divers to participate in an experiment in this manner is not an easy feat—as also mentioned by Godden and Baddeley in some detail—and we fully expected to obtain the same results as these authors. We can only conclude that more replications are necessary of the original experiment, possibly with more subjects, to establish whether the classic study by Godden & Baddeley [1] does indeed deserve the privileged position it currently enjoys as a prime example of context-dependent memory.

Ethics. This study was submitted to the ethics committee and because the research falls within the standard experiments according to the general guidelines of the ethics committee of the University of Amsterdam, Department of Psychology it was, therefore, approved. Details of this are also mentioned in the text of the paper.

Data accessibility. The data are publicly available at https://osf.io/q2vjk/ in Excel format.

Competing interests. I declare I have no competing interests.

Funding. I received no funding for this study.

Acknowledgements. We thank Alan Baddeley for helpful suggestions for designing this replication, diving instructor Nadia Klink for overseeing the safety of the diving sessions, the divers for their willing and enthusiastic participation, and the BlazHoffski crew, in particular Gaelle Blok, for their assistance with experimental procedure and data collection.

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
