## [Peer Review File · Royal Society Open Science]

Review History

RSOS-200724.R0 (Original submission)

Review form: Reviewer 1 (Alan Baddeley)

Do you have any ethical concerns with this paper?

No

Have you any concerns about statistical analyses in this paper?

No

Recommendation?

Reject

Comments to the Author(s)

The experiment proposes to replicate a rather clear example of context-dependent memory whereby material learned in one context is better recalled in the same than in a different context. The effect has been replicated many times using both environmental context, mood context and on drug induced physiological context. The study in to be replicated was remarkable for the size of the effect which was obtained by testing members of a student diving club under two

markedly different environments either beneath 20 feet of water or on a busy beach. The size of the effect is attributed to the extent to the difference between the two environments which in the original study was substantial while in the proposed replication it is much less,, the edge of a hotel swimming pool versus a depth of around six feet. The original study tested one condition per day over four days. The attempted replication carried out everything during one day during which a TV film was being made that included a celebrity participant.

Participants in the original were members of a university diving group with the experiment set up and tested by a fellow diver. In the attempted replication the experimenter was a non-diver and the participants a range of experienced divers present for the day. These latter points might seem trivial but in fact doing experiments beyond the laboratory does introduce a range of important factors including the sample of participants where undergraduate divers tend to be much more homogeneous in age, educational level and background than the general population of diving enthusiasts. The original study tested divers in pairs, the replication in groups of four. I myself have found group testing under diving conditions very challenging, as I explained to Dr Murre, an old friend who discussed the study with me in advance. I suggested at the time said that for all these reasons, together with the distractions of volunteering to carry out an experiment as part of a TV programme with a well known celebrity would all add noise and make it not unlikely that the replication would fail. Hence I would not regard the attempted replication as close, robust or valid.

I do however accept that the broad hypothesis that replication is important , but only if done appropriately.

Hence the basic methodology is sound apart from the identification of the crucial variables which is not simply being underwater but rather the dramatic difference between the two environments in the original study. In the absence of data it is impossible to assess the presence of floor or ceiling effects or indeed the overall comparability and performance between the original study and its attempted replication.

Review form: Reviewer 2 (Sam Berens)

Do you have any ethical concerns with this paper?

No

Have you any concerns about statistical analyses in this paper?

No

Recommendation?

Accept with minor revision

Comments to the Author(s)

This is a well-motivated and timely replication study. Overall, I am enthusiastic about this project but suggest the author includes a few extra details. Most notably, I feel the methods section should include a statistical power/sensitivity analysis which would be crucial when interpreting a null result.

The author states that the very fact Godden and Baddeley's original study is famous provides enough motivation for the study to be replicated. I entirely agree with this claim. However, I also feel it is worthwhile to acknowledge and discuss the theoretical implications that may result from such a replication. While observing the predicted effect may not change much, finding a substantially smaller (or null) effect, may have theoretical implications. Indeed, the context-dependent memory effect is central to numerous models of episodic memory and event segmentation. The author may choose to disregard this advice as it is a minor point that concerns

a personal preference. However, I would recommend that the introduction section is modified to include a brief discussion of possible implications.

I appreciate that the author has chosen to recruit a similar number of participants to the original study. Despite this, I believe the sample size needs to be more thoroughly motivated and discussed in terms of the expected level of statistical power. Please correct me if I am wrong but, based on the original result reported by Godden and Baddeley ($F = 22$, $n = 18$), I calculated an expected effect size of $d = 1.1$. Accounting for the uncertainty in this estimate, I found the replication study ($n = 16$) should yield an expected power of 0.91. Of course, this is more than enough statistical power for a replication. Nonetheless, I feel it is important to explicitly present a power analysis given that the level of a priori power should strongly influence how a null result is interpreted. Furthermore, it is worth noting the smallest effect size that may be reliably detected by this replication, which I calculate to be $d = 0.75$ at a power of 80%. To me, it is important to point out that, while effect sizes below this level may be inconsistent with the original study, much smaller effect sizes (e.g. around 0.2) may still be consistent with models of the context-dependent memory effect.

Decision letter (RSOS-200724.R0)

Dear Dr Murre,

The Editors assigned to your Stage 1 Replication submission ("The Godden and Baddeley (1975) experiment on context-dependent memory on land and underwater: A replication") have now received comments from reviewers. We would like you to revise your paper in accordance with the referee and editors suggestions which can be found below (not including confidential reports to the Editor). Please note this decision does not guarantee eventual acceptance.

Please submit a copy of your revised paper within three weeks (i.e. by the 13-Aug-2020). If deemed necessary by the Editors, your manuscript will be sent back to one or more of the original reviewers for assessment.

When submitting your revised manuscript, you must respond to the comments made by the referees and upload a file "Response to Referees" in the "File Upload" step. Please use this to document how you have responded to the comments, and the adjustments you have made. In order to expedite the processing of the revised manuscript, please be as specific as possible in your response.

Once again, thank you for submitting your manuscript to Royal Society Open Science and I look forward to receiving your revision. If you have any questions at all, please do not hesitate to get in touch. Full author guidelines may be found at <https://royalsocietypublishing.org/rsos/replication-studies#AuthorsGuidance>.

Kind regards,
Professor Chris Chambers
Royal Society Open Science
openscience@royalsociety.org

on behalf of Professor Chris Chambers (Registered Reports Editor, Royal Society Open Science)
openscience@royalsociety.org

Associate Editor Comments to Author (Professor Chris Chambers):

Associate Editor: 1

Comments to the Author:

Two expert reviewers have now assessed the manuscript, and at the outset please accept my apologies the slow handling of your manuscript. This was in no way due to the slowness of the two reviewers who provided assessments (and did so very promptly), but due to other reviewers who accepted the reviewer invitation but then never engaged further and needed to be replaced.

As you will see, the recommendations of the reviewers differ markedly, with Reviewer 1 recommending rejection and Reviewer 2 recommending in-principle acceptance following minor revision. The main point of contention in Reviewer 1's assessment is the concern that the methodology of the replication study deviates in several ways from the methodology of the original study, which is one of the primary criteria for assessing a Stage 1 Replication. I must admit that in reading your manuscript alongside the original study, I shared this concern of the reviewer. For the replication article type, it is important that the methodologies are as close as possible. There is, of course, necessarily some discretion in judging the degree of proximity as no two studies in psychology are ever exactly identical. For this reason, in responding to Reviewer 1's concerns (both in the response to reviewers and in the manuscript), it will be important to provide a convincing argument that these differences are irrelevant to the theory being tested.

Reviewer 2 offers a more positive appraisal and, unlike Reviewer 1, judges that the replication method is sufficiently close to the original study to meet this Stage 1 criterion. The reviewer offers sensible suggestions for considering the potential theoretical implications of the replication in the Introduction and to include a statistical power analysis that takes into account limitations in sensitivity due to sample size.

Comments to Author:

Reviewer: 1

Comments to the Author(s)

The experiment proposes to replicate a rather clear example of context-dependent memory whereby material learned in one context is better recalled in the same than in a different context.

The effect has been replicated many times using both environmental context, mood context and on drug induced physiological context. The study in to be replicated was remarkable for the size of the effect which was obtained by testing members of a student diving club under two markedly different environments either beneath 20 feet of water or on a busy beach. The size of the effect is attributed to the extent to the difference between the two environments which in the original study was substantial while in the proposed replication it is much less,, the edge of a hotel swimming pool versus a depth of around six feet. The original study tested one condition per day over four days. The attempted replication carried out everything during one day during which a TV film was being made that included a celebrity participant.

Participants in the original were members of a university diving group with the experiment set up and tested by a fellow diver. In the attempted replication the experimenter was a non-diver and the participants a range of experienced divers present for the day. These latter points might seem trivial but in fact doing experiments beyond the laboratory does introduce a range of

important factors including the sample of participants where undergraduate divers tend to be much more homogeneous in age, educational level and background than the general population of diving enthusiasts. The original study tested divers in pairs, the replication in groups of four. I myself have found group testing under diving conditions very challenging, as I explained to Dr Murre, an old friend who discussed the study with me in advance. I suggested at the time said that for all these reasons, together with the distractions of volunteering to carry out an experiment as part of a TV programme with a well known celebrity would all add noise and make it not unlikely that the replication would fail. Hence I would not regard the attempted replication as close, robust or valid.

I do however accept that the broad hypothesis that replication is important, but only if done appropriately.

Hence the basic methodology is sound apart from the identification of the crucial variables which is not simply being underwater but rather the dramatic difference between the two environments in the original study. In the absence of data it is impossible to assess the presence of floor or ceiling effects or indeed the overall comparability and performance between the original study and its attempted replication.

Reviewer: 2

Comments to the Author(s)

This is a well-motivated and timely replication study. Overall, I am enthusiastic about this project but suggest the author includes a few extra details. Most notably, I feel the methods section should include a statistical power/sensitivity analysis which would be crucial when interpreting a null result.

The author states that the very fact Godden and Baddeley's original study is famous provides enough motivation for the study to be replicated. I entirely agree with this claim. However, I also feel it is worthwhile to acknowledge and discuss the theoretical implications that may result from such a replication. While observing the predicted effect may not change much, finding a substantially smaller (or null) effect, may have theoretical implications. Indeed, the context-dependent memory effect is central to numerous models of episodic memory and event segmentation. The author may choose to disregard this advice as it is a minor point that concerns a personal preference. However, I would recommend that the introduction section is modified to include a brief discussion of possible implications.

I appreciate that the author has chosen to recruit a similar number of participants to the original study. Despite this, I believe the sample size needs to be more thoroughly motivated and discussed in terms of the expected level of statistical power. Please correct me if I am wrong but, based on the original result reported by Godden and Baddeley ($F = 22$, $n = 18$), I calculated an expected effect size of $d = 1.1$. Accounting for the uncertainty in this estimate, I found the replication study ($n = 16$) should yield an expected power of 0.91. Of course, this is more than enough statistical power for a replication. Nonetheless, I feel it is important to explicitly present a power analysis given that the level of a priori power should strongly influence how a null result is interpreted. Furthermore, it is worth noting the smallest effect size that may be reliably detected by this replication, which I calculate to be $d = 0.75$ at a power of 80%. To me, it is important to point out that, while effect sizes below this level may be inconsistent with the original study, much smaller effect sizes (e.g. around 0.2) may still be consistent with models of the context-dependent memory effect.

Author's Response to Decision Letter for (RSOS-200724.R0)

See Appendix A.

Decision letter (RSOS-200724.R1)

Dear Dr Murre

On behalf of the Editor, I am pleased to inform you that your Manuscript RSOS-200724.R1 entitled "The Godden and Baddeley (1975) experiment on context-dependent memory on land and underwater: A replication" has been accepted in principle for publication in Royal Society Open Science.

You may now progress to Stage 2 and complete the study as approved.

Please note that you must now register your approved protocol on the Open Science Framework (<https://osf.io/rr>), using the 'Submit your approved Registered Report' option and then the 'Registered Report Protocol Preregistration' option. Please use the Registered Report option even though your article is being accepted as a Stage 1 Replication. Further into the registration process, in the Journal Title field enter 'Royal Society Open Science (Replication article type, Results-Blind track)'. Please note that a time-stamped, independent registration of the protocol is mandatory under journal policy, and manuscripts that do not conform to this requirement cannot be considered at Stage 2. The protocol should be registered unchanged from its current approved state. Please include a URL to the protocol in your Stage 2 manuscript, and because you submitted via the Results-Blind track please note in the manuscript that the pre-registration was performed after data analysis (e.g. 'This article received results-blind in-principle acceptance (IPA) at Royal Society Open Science. Following IPA, the accepted Stage 1 version of the manuscript, not including results and discussion, was preregistered on the OSF (URL). This preregistration was performed after data analysis.')

We would be grateful if you could now update the journal office as to the anticipated completion date of your study.

Following completion of your study, we invite you to resubmit your paper for peer review as a Stage 2 Replication. Please note that your manuscript can still be rejected for publication at Stage 2 if the Editors consider any of the following conditions to be met:

- The Introduction and methods deviated from the approved Stage 1 submission (required).
- The authors' conclusions were not considered justified given the data.

We encourage you to read the complete guidelines for authors concerning Stage 2 submissions at: <https://royalsocietypublishing.org/rsos/replication-studies#AuthorsGuidance>. Please especially note the requirements for data sharing and that withdrawing your manuscript will result in publication of a Withdrawn Registration.

Once again, thank you for submitting your manuscript to Royal Society Open Science and I look forward to receiving your Stage 2 submission. If you have any questions at all, please do not hesitate to get in touch. We look forward to hearing from you shortly with the anticipated submission date for your stage two manuscript.

Kind regards,
Professor Chris Chambers

Author's Response to Decision Letter for (RSOS-200724.R1)

See Appendix B.

RSOS-200724.R2 (Revision)

Review form: Reviewer 1 (Alan Baddeley)

Is the manuscript scientifically sound in its present form?

No

Is the language acceptable?

Yes

Do you have any ethical concerns with this paper?

Yes

Have you any concerns about statistical analyses in this paper?

No

Recommendation?

Reject

Comments to the Author(s)

Please send the author a copy of the comments to the editor.

Review form: Reviewer 2 (Sam Berens)

Is the manuscript scientifically sound in its present form?

Yes

Is the language acceptable?

Yes

Do you have any ethical concerns with this paper?

No

Have you any concerns about statistical analyses in this paper?

No

Recommendation?

Accept with minor revision

Comments to the Author(s)

I believe the submission meets all the Stage 2 criteria for a Replication Study. I only have a few minor points.

I cannot access the Excel data sheet with the OSF link that has been provided; I believe that the document has yet to be made public.

There may be something amiss with how the sensitivity analysis is reported in the Subjects sub-section. Specifically, while the computed value of EtaSquared appears to be correct ($\eta^2 = 0.647$), the derivation of this statistic should be as follows...

$$\eta^2 = F / (F + N - k)$$

$$\eta^2 = 22 / (22 + 16 - 4)$$

Where 'N-k' ($16 - 4 = 12$) is the error degrees of freedom ('df2') reported by Godden and Baddeley. I would suggest using 'df2' to denote this term as it is easier to understand and does not require defining 'k'. Furthermore, I don't understand how the author arrived at a value of 1.68 for a lower bound value of Cohen's d. I may be missing something here but it appears that the expected effect size should be $d = 2.71$ so how was this used to get a lower bound?

I think that the following statement is incorrect or properly qualified: "Thus, replicating the original design exactly as is, would yield a null result with $p < 0.05$ given that the real effect was the same." Would something like this be more appropriate?: "Thus, replicating the original design exactly as is, should yield more than 99.9% statistical power assuming the lower bound effect size is accurate."

I recommend explicitly stating both df1 and df2 for each reported F-test to avoid ambiguity.

The ethics statement is rather long-winded and difficult to understand. Perhaps consider rewording.

The sentence beginning "A difference with ..." in the Subjects sub-section does not appear to be grammatically correct.

Decision letter (RSOS-200724.R2)

Dear Dr Murre

On behalf of the Editor, I am pleased to inform you that your Stage 2 Replication submission RSOS-200724.R2 entitled "The Godden and Baddeley (1975) experiment on context-dependent memory on land and underwater: A replication" has been accepted for publication in Royal Society Open Science subject to minor revision in accordance with the referee suggestions. Please find the referees' comments at the end of this email.

The reviewers and Subject Editor have recommended publication, but also suggest some minor revisions to your manuscript. Therefore, I invite you to respond to the comments and revise your manuscript.

Please also ensure that all the below editorial sections are included where appropriate (a non-exhaustive example is included in an attachment):

- Ethics statement

- Data accessibility

If you wish to submit your supporting data or code to Dryad (<http://datadryad.org/>), or modify your current submission to dryad, please use the following link:
<http://datadryad.org/submit?journalID=RSOS&manu=RSOS-200724.R2>

- Competing interests

- Authors' contributions

- Acknowledgements

- Funding statement

Because the schedule for publication is very tight, it is a condition of publication that you submit the revised version of your manuscript within 7 days (i.e. by the 29-Jul-2021). If you do not think you will be able to meet this date please let me know immediately.

- 1) A text file of the manuscript (tex, txt, rtf, docx or doc), references, tables (including captions) and figure captions. Do not upload a PDF as your "Main Document".
- 2) A separate electronic file of each figure (EPS or print-quality PDF preferred (either format should be produced directly from original creation package), or original software format)
- 3) Included a 100 word media summary of your paper when requested at submission. Please ensure you have entered correct contact details (email, institution and telephone) in your user account
- 4) Included the raw data to support the claims made in your paper. You can either include your data as electronic supplementary material or upload to a repository and include the relevant DOI within your manuscript
- 5) Included your supplementary files in a format you are happy with (no line numbers, Vancouver referencing, track changes removed etc) as these files will NOT be edited in production

Kind regards,
Professor Chris Chambers
Royal Society Open Science
openscience@royalsociety.org

Associate Editor Comments to Author (Professor Chris Chambers):
Comments to the Author:

The two reviewers who assessed the Stage 1 manuscript kindly returned to evaluate the Stage 2 submission. As you will see, their recommendations differ, with Reviewer 1 remaining negative and Reviewer 2 noting relatively minor points of clarification. As regards the concerns raised by Reviewer 1, most of which overlap with the reviewer's Stage 1 evaluation, where any of these concerns have not been addressed already in the Discussion (which I note several have already) please do so accordingly. It is notable that the reviewer judges that "the level of control over extraneous variables together with deviation from the original make this study scientifically unacceptable." Given that the review history for this submission will be public, I should note for the record that while I agree with the reviewer's general point that your replication attempt

deviated from the original study, I do not agree that the deviations are sufficient to negate the scientific value of the work or be so far removed from the original study as to fall outside the scope of the RSOS Replications format. I have therefore reached an editorial decision at variance with the recommendation of this reviewer.

Provided you are able to respond comprehensively to these reviewers, final Stage 2 acceptance should be forthcoming without requiring further in-depth review.

Reviewers' comments to Author:

Reviewer: 1

Comments to the Author(s)

As the author points out, the occurrence of context dependency is well established. The Godden and Baddeley study is unusual in demonstrating it particularly clearly by achieving two radically different physical contexts while minimising other distractions. Such potential distractions are likely to be substantial when, as in the proposed replication, a group of strangers are tested in groups of four as part of an opportunity to appear on a television programme alongside a celebrity who is describing the proceedings. Good psychological experiments depend crucially participants not being distracted by extraneous factors such as how they will appear on a TV show.

The situation is further constrained by the need to pack a four day study into an hour and a half and use members of the TV crew as research assistants.

There are in addition many differences between the two studies including:

The design which tries to combine the initial two experiments into one, adding potential confounding.

2. The context dependency effect operates through a clear contrast between the two environment.

This does not simply reflect whether the participant is tested above or below the water surface.

Hence a depth of 6 feet in a heated hotel swimming pool is not the same as 20 feet in a cold opensea environment, a well-established difference (e.g. Baddeley, 1966, *J. App Psych*, 81-85)

4. The delivery of stimuli is crucial. In the original study, individuals had direct phone contact, and were trained beforehand to control their breathing patterns to avoid sound masking. The study relied on a single underwater speaker for the four divers with no specific training. This is probably the cause of the underwater deficit observed in this but not the earlier study.

In short, while I accept that a well controlled replication would be desirable, the level of control over extraneous variables together with deviation from the original make this study scientifically unacceptable.

Reviewer: 2

Comments to the Author(s)

I believe the submission meets all the Stage 2 criteria for a Replication Study. I only have a few minor points.

I cannot access the Excel data sheet with the OSF link that has been provided; I believe that the document has yet to be made public.

There may be something amiss with how the sensitivity analysis is reported in the Subjects subsection. Specifically, while the computed value of EtaSquared appears to be correct ($\eta^2 = 0.647$), the derivation of this statistic should be as follows...

$$\eta^2 = F / (F + N - k)$$

$$\eta^2 = 22 / (22 + 16 - 4)$$

Where 'N-k' (16-4 = 12) is the error degrees of freedom ('df2') reported by Godden and Baddeley. I would suggest using 'df2' to denote this term as it is easier to understand and does not require defining 'k'. Furthermore, I don't understand how the author arrived at a value of 1.68 for a lower

bound value of Cohen's d . I may be missing something here but it appears that the expected effect size should be $d = 2.71$ so how was this used to get a lower bound?

I think that the following statement is incorrect or properly qualified: "Thus, replicating the original design exactly as is, would yield a null result with $p < 0.05$ given that the real effect was the same." Would something like this be more appropriate?: "Thus, replicating the original design exactly as is, should yield more than 99.9% statistical power assuming the lower bound effect size is accurate."

I recommend explicitly stating both df_1 and df_2 for each reported F-test to avoid ambiguity.

The ethics statement is rather long-winded and difficult to understand. Perhaps consider rewording.

The sentence beginning "A difference with ..." in the Subjects sub-section does not appear to be grammatically correct.

Author's Response to Decision Letter for (RSOS-200724.R2)

See Appendix C.

Decision letter (RSOS-200724.R3)

Dear Dr Murre:

It is a pleasure to accept your manuscript entitled "The Godden and Baddeley (1975) experiment on context-dependent memory on land and underwater: A replication" in its current form for publication in Royal Society Open Science.

on behalf of Professor Chris Chambers (Subject Editor)
openscience@royalsociety.org

Appendix A

Amsterdam, 11 March 2021

Dear Professor Chambers,

Please find enclosed the revised version of my submission to the *Royal Society Open Science*, originally entitled 'The Godden and Baddeley (1975) experiment on context-dependent memory on land and underwater: A replication'. It was submitted as a *Stage 1 manuscript as a Replication article*.

Let me first apologize for the long delay in revising this article and thank you for the extension of the deadline.

I am also grateful to the reviewers, who make many excellent points. Below, I will address each one in turn with pointers to where and how the text has been updated.

Sincerely,

Jaap Murre

*Prof.dr. Jaap Murre
Department of Psychology
University of Amsterdam*

Reviewer's concerns and my actions

Reviewer 1

Reviewer 1 makes several excellent points:

The size of the effect is attributed to the extent to the difference between the two environments which in the original study was substantial while in the proposed replication it is much less...

This is true and already mentioned already right at the beginning of the Methods section. It is also very much a valid point, which we emphasize in the Discussion.

the sample of participants where undergraduate divers tend to be much more homogeneous in age, educational level and background than the general population of diving enthusiasts.

Again, a good point. We have added a sentence to the Subjects section to emphasize this point.

The original study tested divers in pairs, the replication in groups of four. I myself have found group testing under diving conditions very challenging, as I explained to Dr Murre, an old friend who discussed the study with me in advance. I suggested at the time said that for all these reasons, together with the distractions of volunteering to carry out an experiment as part of a TV programme with a well known celebrity would all add noise and make it not unlikely that the replication would fail. Hence I would not regard the attempted replication as close, robust or valid.

I agree with Reviewer 1 here that the crew and celebrity may have added some noise to the experiment. I do not agree with his conclusion that therefore the replication attempt is not 'close, robust or valid'. The camera crew was mainly in the background, the celebrity (whose data were not analyzed) did a few camera outtakes away from the other subjects, but only after a specific step in the procedure had been fully completed. Also, there were severe distractions in the original experiment (e.g., one subject almost being run over by a DUKW). Part of the strength of the original study is that it does find an effect under relatively naturalistic—hence quite noisy—conditions. We do discuss all of these concerns in the Discussion.

In the absence of data^a it is impossible to assess the presence of floor or ceiling effects^b or indeed the overall comparability and performance^c between the original study and its attempted replication.

a. As is clearly stated in the paper, the raw data were and remain available at URL <https://osf.io/q2vjk/>.

b. There are no ceiling effects. Far from it: The highest score was a recall of 19 out of 36 possible words on a list. All scores were much lower than 36.

c. I do believe the results are roughly comparable, though the subjects in the replication attempt scored on the whole somewhat lower than in the original experiment. This is, perhaps, because the original subjects were younger (average age is not given in the original study but as undergraduates would be around 20) and higher educated (which is associated with higher scores on similar [neuropsychological] tests, e.g., the AVLT). The original study does not mention any details about the characteristics of the words used, so they may have been easier to remember. These concerns are now mentioned in the Discussion.

Reviewer 2

Reviewer 2 is positive about the paper and suggests a few modifications:

the methods section should include a statistical power/sensitivity analysis which would be crucial when interpreting a null result. Reviewer 2 also suggests we discuss the implications of (possible lack of) power in the paper in more detail.

We have included a thorough discussion of effect sizes sample size, and probability of finding a null result in the Method section (see Subjects).

There is one remark, we must make where the reviewer is not correct: Reviewer 2 calculates an effect size of Cohen's $d = 1.1$ with $d=0.9$ in the replication, assuming 18 subjects in the original experiment and 16 in the replication. This is not correct, the original study had 16 subjects, two of which were not analyzed due to technical difficulties. They were replaced by two new ones who were tested in a different location ('freshwater site'). So, both experiments analyzed 16 subjects and, importantly, our replication does not have fewer subjects. Also, we opted to calculate Cohen's d via eta-squared, rather than use the reviewer's calculation, which we could not replicate (see Subjects section).

it is worthwhile to acknowledge and discuss the theoretical implications that may result from such a replication ... I would recommend that the introduction section is modified to include a brief discussion of possible implications

Reviewer 2 makes a good suggestion here and I indeed started writing this but then opted against it. There is a lot of debate about possible mechanisms involved in context-dependent memory and there are many experimental studies that sometimes do and sometimes do not find effects (I already mentioned the weak Cohen's d of 0.25 for the experiments in this field). These could easily fill two or three pages (of the MS Word file) and would in my opinion detract from the main subject of the paper: a simple replication study. Also, finding a null result, as I do here, almost always require additional replication attempts before one can decide whether there really is or isn't an effect. As such, it has no direct implications but rather places a 'red flag'.

Appendix B

Amsterdam, 8 June 2021

Dear Professor Chambers,

Please find enclosed the revised version of my submission to the *Royal Society Open Science*, originally entitled 'The Godden and Baddeley (1975) experiment on context-dependent memory on land and underwater: A replication'. It was submitted as a *Stage 1 manuscript as a Replication article*, after which I received an invitation to proceed to *Stage 2*. I

I have registered the protocol on the Open Science Foundation as requested, with DOI 10.17605/OSF.IO/KFHVU and OSF URL <https://osf.io/kfhvu>.

Sincerely,

Jaap Murre

*Prof.dr. Jaap Murre
Department of Psychology
University of Amsterdam*

Appendix C

Amsterdam, 7 October 2021

Dear Professor Chambers,

Let me first apologize for not meeting the original deadline of 29th of July 2021; I am grateful for the opportunity to submit my revision after all.

Please find enclosed the revised final version of my submission to the *Royal Society Open Science*, originally entitled 'The Godden and Baddeley (1975) experiment on context-dependent memory on land and underwater: A replication'.

Below, I will address the final points raised by you and the reviewers with pointers to where and how the text has been updated.

Sincerely,

Jaap Murre

*Prof.dr. Jaap Murre
Department of Psychology
University of Amsterdam*

Point-by-point overview of requested changes

General points raised in the decision letter

Ethics

This has been in the text from the beginning. See page 6:

Medical-ethical approval was sought and obtained from the Medical-Ethical Committee of the University of Amsterdam (working title in committee registry: 'Geheugen Onder Water').

But on request of Reviewer 2, this has now been changed to:

Ethical approval was obtained from the Medical-Ethical Committee of the University of Amsterdam (title in the registry: 'Geheugen Onder Water').

Note that *Medical-Ethical Committee* is an official (legal) term in the Netherlands and should not be abbreviated to Ethics Committee.

Data accessibility

Reviewer 2 was correct: The data are indeed at the Open Science Foundation, at <https://osf.io/q2vjk/>, but I had forgotten to publish the entire item, just the data files. This has now been remedied, so that **it is now possible to easily see the description of the project and download the data** without logging in to the system.

Competing interests

I now declare in the text on page 5 that I have no competing interests.

Authors' contributions

I am the only author. No other researchers contributed significantly to the substance of this work.

Acknowledgements

These were already up to date.

Funding statement

I have now included a line on page 5:

The study was funded by the University of Amsterdam as part of my normal work as a researcher.

Comments by Reviewer 1

The Godden and Baddeley study is unusual in demonstrating it particularly clearly by achieving two radically different physical contexts while minimising other distractions. Such potential distractions are likely to be substantial when, as in the proposed replication, a group of strangers are tested in groups of four as part of an opportunity to appear on a television programme alongside a celebrity who is describing the proceedings. Good psychological experiments depend crucially participants not being distracted by extraneous factors such as how they will appear on a TV show.

The situation is further constrained by the need to pack a four day study into an hour and a half and use members of the TV crew as research assistants.

This reiterates earlier criticism by Reviewer 1; these differences between the replication and the original are acknowledged as valid points in the Discussion (p. 11).

**There are in addition many differences between the two studies including:
The design which tries to combine the initial two experiments into one, adding potential confounding.**

This is also discussed at several points in the paper. I honestly believe that the first part of the original study was affected by a confound between 'land-land and water-water' and lack of activity (which is present in land-water and water-land) and that the second part of the original experiment remedied that limitation; it was therefore only that part that was worthy of replication.

The context dependency effect operates through a clear contrast between the two environment. This does not simply reflect whether the participant is tested above or below the water surface. Hence a depth of 6 feet in a heated hotel swimming pool is not the same as 20 feet in a cold opensea environment , a well-established difference (e.g. Baddeley,1966, J. App Psych,81-85)

This is again a valid point that may warrant additional replication attempts that remedy that, which is also acknowledged as such in the Discussion (p.11).

The delivery of stimuli is crucial. In the original study, individuals had direct phone contact , and were trained beforehand to control their breathing patterns to avoid sound masking. The study relied on a single underwater speaker for the four divers with no specific training. This is probably the cause of the underwater deficit observed in this but not the earlier study.

As is described in the Procedure on p.7 and 8, the divers in each group were instructed in breathing and listening and did then practice the breathing patterns on land and under water. E.g., on p. 8:

While they were underwater, two test runs with the underwater speakers were run, the second of which consisted of the experimenter (JM) reading a list of 36 words similar to the experimental lists.

In short, while I accept that a well controlled replication would be desirable, the level of control over extraneous variables together with deviation from the original make this study scientifically unacceptable.

I am grateful that the Editor chose to disagree with this statement.

Comments by Reviewer 2

I cannot access the Excel data sheet with the OSF link that has been provided; I believe that the document has yet to be made public.

This is a valid point, which has now been remedied. See the General Points above.

There may be something amiss with how the sensitivity analysis is reported in the Subjects sub-

section. Specifically, while the computed value of EtaSquared appears to be correct ($\eta^2 = 0.647$), the derivation of this statistic should be as follows...

$$\eta^2 = F/(F+N-k)$$

$$\eta^2 = 22/(22+16-4)$$

Where 'N-k' ($16-4 = 12$) is the error degrees of freedom ('df2') reported by Godden and Baddeley. I would suggest using 'df2' to denote this term as it is easier to understand and does not require defining 'k'. Furthermore, I don't understand how the author arrived at a value of 1.68 for a lower bound value of Cohen's *d*. I may be missing something here but it appears that the expected effect size should be $d = 2.71$ so how was this used to get a lower bound?

I am very grateful to the reviewer for pointing out these typos. I have also followed the suggestion to replace N-k by *df2* in the text. I recalculated *d* and agree with Reviewer's 1 suggested correction. These numbers are now updated in the text.

I think that the following statement is incorrect or properly qualified: "Thus, replicating the original design exactly as is, would yield a null result with $p < 0.05$ given that the real effect was the same." Would something like this be more appropriate?: "Thus, replicating the original design exactly as is, should yield more than 99.9% statistical power assuming the lower bound effect size is accurate."

I have followed this suggestion and also updated the following sentence to reflect this (see p. 7):

Thus, replicating the original design exactly as is, should yield more than 99.9% statistical power assuming the lower bound effect size is accurate. Hence, there is a very low chance of finding a spurious null result.

I recommend explicitly stating both df1 and df2 for each reported F-test to avoid ambiguity.

This has now been done (three occurrences on page 9).

The ethics statement is rather long-winded and difficult to understand. Perhaps consider rewording.

This text has now been tightened (also see *Ethics* above, under *General Points*).

The sentence beginning "A difference with ..." in the Subjects sub-section does not appear to be grammatically correct.

This indeed had a typo, which has now been corrected.